# Sensitivity and Specificity of Patient-Reported Clinical Manifestations to Diagnose COVID-19 in Adults from a National Database in Chile: A Cross-Sectional Study

**DOI:** 10.3390/biology11081136

**Published:** 2022-07-29

**Authors:** Felipe Martinez, Sergio Muñoz, Camilo Guerrero-Nancuante, Carla Taramasco

**Affiliations:** 1Unidad de Cuidados Intensivos, Hospital Naval Almirante Nef, Viña del Mar 2520000, Chile; 2Facultad de Medicina, Escuela de Medicina, Universidad Andrés Bello, Viña del Mar 2531015, Chile; 3Concentra Educación e Investigación Biomédica, Viña del Mar 2552906, Chile; 4Departamento de Salud Pública-CIGES, Facultad de Medicina, Universidad de La Frontera, Francisco Salazar 1145, Temuco 4811230, Chile; sergio.munoz.n@ufrontera.cl; 5Escuela de Enfermería, Universidad de Valparaíso, Valparaíso 2500000, Chile; camilo.guerrero@uv.cl; 6Facultad de Ingeniería, Universidad Andrés Bello, Millennium Nucleus on Sociomedicine, Viña del Mar 2520000, Chile; carla.taramasco@unab.cl

**Keywords:** COVID-19, clinical manifestations, diagnostic accuracy, risk factors

## Abstract

**Simple Summary:**

COVID-19 is frequently suspected based on clinical features, such as fever, cough, headache, or loss of taste. However, it remains unclear whether these manifestations are reliable indicators of disease. We sought to evaluate the diagnostic accuracy of clinical manifestations in identifying patients with COVID-19. Data from a nationwide database comprising of 2,187,962 patients who sought medical care in Chile were analysed. Information regarding age, gender, type of insurance, a history of a close contact with COVID-19, and several clinical features was obtained. The most common complaints were headache, muscle aches, and cough. No single clinical feature was precise enough to fully confirm or exclude COVID-19. The combination of several of these manifestations with epidemiological risk factors into a model showed a reasonable accuracy in detecting cases of COVID-19.

**Abstract:**

(1) Background: The diagnosis of COVID-19 is frequently made on the basis of a suggestive clinical history and the detection of SARS-CoV-2 RNA in respiratory secretions. However, the diagnostic accuracy of clinical features is unknown. (2) Objective: To assess the diagnostic accuracy of patient-reported clinical manifestations to identify cases of COVID-19. (3) Methodology: Cross-sectional study using data from a national registry in Chile. Infection by SARS-CoV-2 was confirmed using RT-PCR in all cases. Anonymised information regarding demographic characteristics and clinical features were assessed using sensitivity, specificity, and diagnostic odds ratios. A multivariable logistic regression model was constructed to combine epidemiological risk factors and clinical features. (4) Results: A total of 2,187,962 observations were available for analyses. Male participants had a mean age of 43.1 ± 17.5 years. The most common complaints within the study were headache (39%), myalgia (32.7%), cough (31.6%), and sore throat (25.7%). The most sensitive features of disease were headache, myalgia, and cough, and the most specific were anosmia and dysgeusia/ageusia. A multivariable model showed a fair diagnostic accuracy, with a ROC AUC of 0.744 (95% CI 0.743–0.746). (5) Discussion: No single clinical feature was able to fully confirm or exclude an infection by SARS-CoV-2. The combination of several demographic and clinical factors had a fair diagnostic accuracy in identifying patients with the disease. This model can help clinicians tailor the probability of COVID-19 and select diagnostic tests appropriate to their setting.

## 1. Introduction

In December 2019, a novel respiratory disease named COVID-19 was reported in the city of Wuhan, China [1,2]. The disease occurs due to infection with a novel coronavirus (SARS-CoV-2), which in some cases can induce respiratory failure, multiorgan dysfunction, and death [3]. In early January 2020, the virus was sequenced, allowing for the development of diagnostic testing based on the detection of viral nucleic acid, such as real-time reverse transcription polymerase chain reaction (RT-PCR). As of October 2021, the disease had affected more than 240 million people worldwide, causing more than 4.5 million deaths [4]. The rapid spread of the disease has been explained by the high transmissibility of the virus, especially amongst new variants such as delta and omicron, the apparent absence of any cross-protective immunity from related viral infections, and delayed public health response measures [5,6,7,8,9]. Several health institutions reported shortages of required materials for pharyngeal specimen collection, sample extraction, and RT-PCR materials [10]. This led to a recommendation to urgently develop alternative methods to diagnose the disease, especially in countries where the growth of the pandemic had rapidly outpaced the capacity to test for it.

Several reviews have suggested that the diagnosis of COVID-19 can be established on the basis of a suggestive clinical history and the detection of SARS-CoV-2 RNA in respiratory secretions [1,11]. Observational reports have been heterogeneous regarding the frequency of specific manifestations of the disease. Symptoms such as fever, dry cough, shortness of breath, fatigue, nausea/vomiting, diarrhoea, and myalgia are common amongst hospitalised patients [2,12,13]. Anosmia and gustatory dysfunctions (dysgeusia or ageusia) have been also cited as standalone markers of disease [14,15,16]. However, it should be considered that many of these disease manifestations are of a non-specific nature, and other reports have pointed out that their frequency might be overestimated due to a high proportion of oligosymptomatic patients [17,18]. Given the uncertainty regarding the diagnostic accuracy of clinical features to identify patients with COVID-19, we conducted this study to address this issue.

## 2. Methodology

A cross-sectional study was undertaken using data available in a national registry in Chile from March 2020 to January 2021. This registry holds data of each patient that sought medical attention due to symptoms of COVID-19 and of persons that were screened for the disease within national territories. The study protocol was approved by the Ethics Committee of the Faculty of Medicine at the University of Valparaiso. Its approval number is 15-2020. 

### 2.1. Participants

Eligible participants included adult patients (>18 years of age) who sought medical attention due to clinical complaints in whom COVID-19 was suspected by the attending physician, irrespective of their health insurance. Patients were consecutively enrolled within the database. Those who were asymptomatic and were diagnosed by active surveillance campaigns or close-contact screening programmes were excluded. 

A basic clinical profile for every included participant was obtained from a registry named EPIVIGILA (Trademark N° 2020-A-7079) [19]. This registry has been endorsed by the Chilean Ministry of Health to hold every suspected and confirmed case of COVID-19 in Chile. This database holds information from the first patient that was diagnosed with a SARS-CoV-2 infection within national boundaries and is currently used by the Ministry of Health to monitor the pandemic course in Chile. Notification of suspected cases is mandated by law within 24 hours of first contact with a physician. The registry includes data on sex, age, nationality, current residence, recent travel, occupation, clinical features, context of testing (active screening vs. spontaneous consultation) and several comorbidities. In addition, contacts of confirmed cases of COVID-19 are paired within the database, thus facilitating surveillance of the pandemic. All data were prospectively collected during the patient’s initial contact with the healthcare provider.

### 2.2. Clinical Features

Relevant clinical manifestations included fever, cough, headache, malaise, sore throat, chest pain, abdominal pain, diarrhoea, dyspnea, cyanosis, tachypnea, dysgeusia/ageusia, anosmia, prostration, and myalgia. These manifestations were self-reported by the patient at first consult, or a proxy in cases where a detailed clinical history was not obtainable (i.e., dementia, delirium, or other forms of disturbances in cognition). No standardised procedure for obtaining these data was used within the registry. Clinical information was not made available for technicians performing RT-PCR tests.

### 2.3. Reference Standard

National guidelines have established RT-PCR tests from respiratory samples (i.e., nasopharyngeal swabs) as the reference standard to confirm COVID-19. Therefore, we used this test as a reference standard in our study. However, it has also been recognised that several factors might modify this test’s diagnostic performance, including sample collection issues and time from first exposure. To minimise diagnostic errors, the Ministry of Health established national standards for public and private laboratories conducting RT-PCR tests early within the course of the pandemic. Centres were required to obtain certification to offer these tests to the community, and RT-PCR tests registered within EPIVIGILA had to have been conducted in one of these centres. A second RT-PCR test was conducted amongst patients with indeterminate test results and then classified, based on the second test, as positives or negatives. Cases with an indeterminate result in the second test were excluded from analyses, but the number of events was registered within the database. 

### 2.4. Statistical Analyses

Descriptive statistics, including medians, means, standard deviations, interquartile ranges (IQR), and absolute and relative frequencies, were used first to describe the characteristics of the study participants. Bivariate comparisons between groups were performed using the Chi-square test for categorical variables and Student’s *t* test for continuous variables. The observed diagnostic accuracy was summarised using sensitivity, specificity, diagnostic odds ratios and both positive and negative likelihood ratios. Ninety-five percent confidence intervals were calculated for these estimators as well. In addition to bivariate analyses, multivariable logistic regression was used to combine individual clinical manifestations with the highest diagnostic accuracy in bivariate comparisons. Briefly, these models considered any variable that showed a two-tailed p-value of less than 0.15 as a candidate variable to be included in a multivariable index. All candidate models considered the possibility of interactions between independent variables during their construction. In order to control for potential differences in sociodemographic factors, data on gender, age, and health insurance were kept for all candidate models. The diagnostic accuracy of individual models was quantified using receiver–operator characteristics (ROC) curves. These curves were, in turn, compared using 95% confidence intervals. Overall goodness of fit was assessed using Hosmer and Lemeshow’s statistic. Analyses were undertaken by an independent statistician using anonymised data with STATA 17.0 MP^®^ (StataCorp LLC, College Station, TX, USA). A two-sided *p*-value of less than 5% was considered to be statistically significant. 

## 3. Results

### 3.1. Participant Characteristics

The database contained a total of 5,789,289 cases (Figure 1). Of these, 3,291,135 belonged to active screening programmes, which left 2,498,154 eligible observations for analyses. A total of 310,192 samples belonged to patients younger than 18 years of age and 17,428 (0.53%) patients received an initial indeterminate RT-PCR result. In most cases, COVID-19 was excluded in the second test (73.0%), but 101 patients continued to show indeterminate results. This left 2,187,962 patients available for analyses.

Male participants had a mean age of 43.1 ± 17.5 years. Ninety-two percent were Chilean and 2.7% belonged to an indigenous Chilean population. Most patients resided within the Metropolitan region in central Chile (39.8%). The most common comorbidities were arterial hypertension (11.7%), which was followed by diabetes mellitus (5.9%) and asthma (2.8%). A history of recent travel was described by 1.93% of included participants. International travel was the most frequently reported (1.18%). A close contact was reported by 13% of the study participants, and 10.5% corresponded to confirmed cases of COVID-19.

Due to the large sample size, statistical tests showed significant differences in almost every clinical and sociodemographic characteristic when patients with positive SARS-CoV-2 RT-PCR were contrasted with those in whom the disease was not confirmed. Patients with confirmed COVID-19 tended to have more comorbidities, including arterial hypertension (15.2 vs. 11.2%, *p* < 0.001) and diabetes mellitus (8.3 vs. 5.5%, *p* < 0.001); a confirmed contact was more frequently described in this group as well (18.3 vs. 9.3%, *p* < 0.001). The frequency of suspected cases of COVID-19 was similar between study groups (2.29 vs. 2.01%, *p* < 0.001). A higher frequency of indigenous Chilean populations was noted in this group as well (3.9 vs. 2.5%, *p* < 0.001). National and international travel was less common amongst patients with confirmed COVID-19 (0.28 vs. 1.3% and 0.35% vs. 0.81%, *p* < 0.001 for both comparisons). A complete description of patient characteristics is shown in Table 1.

### 3.2. Diagnostic Accuracy of Clinical Features

The most common complaints within the study were headache (39%), myalgia (32.7%), cough (31.6%), and sore throat (25.7%). Dyspnea or tachypnea were uncommon presenting symptoms, with a relative frequency of 8.7 and 1.2%, respectively. Anosmia or dysgeusia/ageusia were rarely described by included participants, with an overall prevalence of 5.0 and 4.1%, respectively. Gastrointestinal symptoms such as diarrhoea or abdominal pain were uncommon as well, with relative frequencies of 7.0 and 8.9%. 

Overall, most clinical features tended to have a low diagnostic accuracy for differentiating COVID-19 from other clinical entities. The most sensitive disease manifestation was headache, with a sensitivity of 56.5% (95% CI 56.4–56.7%), and was followed by myalgia (sensitivity of 53.3%, 95% CI 53.1–53.5%) and cough (sensitivity of 51.1%, 95% CI 51.0–51.3%). On the other hand, the most specific clinical features were cyanosis (specificity of 99%, 95% CI 99.8–99.9%), tachypnea (specificity of 98.9%, 95% CI 98.9–98.9%), anosmia (specificity of 97.5%, 95% CI 97.5–97.5%), and dysgeusia/ageusia (specificity of 97.0%, 95% CI 97.0–97.0%). The latter two manifestations also showed the highest diagnostic odds ratios, with 6.95 (95% CI 6.86–7.04) and 6.19 (95% CI 6.11–6.28), and were followed by myalgia (2.74, 95% CI 2.71–2.76), cough (2.62, 95% CI 2.60–2.64), fever (2.34, 95% CI 2.88–2.92), and headache (2.30, 95% CI 2.27–2.31). A complete description of the observed diagnostic accuracy of these clinical manifestations is shown in Table 2.

### 3.3. Multivariable Models

In order to establish a pre-test probability of COVID-19, a model was constructed using a multivariate logistic regression. The full model consisted of 21 variables that included every clinical manifestation and several epidemiological characteristics of the study sample. When these features were combined, the most relevant elements that increased the overall probability of confirming an infection by SARS-CoV-2 were a history of a confirmed contact with COVID-19 (aOR 2.27, 95% CI 2.24–2.30, *p* < 0.001), anosmia (aOR 3.72, 95% CI 3.66–3.79, *p* < 0.001), fever (aOR 2.23, 95% CI 2.21–2.24, *p* < 0.001), myalgia (aOR 1.68, 95% CI 1.67–1.70, *p* < 0.001), cough (aOR 1.88, 95% CI 1.86–1.90, *p* < 0.001), and dysgeusia/ageusia (aOR 1.81, IC95% 1.77–1.84, *p* < 0.001). The most important clinical feature that reduced the probability of COVID-19 was abdominal pain (aOR 0.61, 95% CI 0.60–0.62, *p* < 0.001). The complete model is shown in Table 3, and its overall accuracy was fair, with an area under the ROC curve of 0.753 (95% CI 0.752–0.754).

In order to facilitate the clinical use of these equations, a simplified model that considered four demographic characteristics (age, sex, type of insurance (private vs. other), and a history of belonging to an indigenous Chilean population) and the six most important clinical manifestations was estimated as well. These factors tended to maintain their strength of association that was calculated in the full model, as shown in Table 4. This simplified model showed a similar diagnostic accuracy when contrasted to the full model, with an AUC of 0.744 (95% CI 0.743–0.746). The comparative accuracy of both models is shown in Figure 2.

## 4. Discussion

This is the largest cross-sectional study addressing risk factors and frequently reported clinical manifestations of disease regarding their diagnostic accuracy to identify patients infected with SARS-CoV-2. One of the key findings is that no single clinical feature showed sufficient precision to be deemed a standalone marker of disease, which is in agreement with the results from a recent systematic review [20]. This highlights the need to combine clinical features with epidemiological risk factors in order to improve the evaluation of a pre-test probability of COVID-19. Our data further accentuates the relevance of active surveillance programmes to obtain reliable estimates of the pandemic burden, given the apparent lack of accuracy of clinical symptoms and the significant proportion of cases that were detected by screening strategies.

The combination of several epidemiological and clinical elements had a fair discrimination capacity to identify patients with COVID-19. Although the 95% confidence intervals did not overlap because of the large sample size, the reduction in the number of variables to contain four key demographic characteristics and the six diagnostic features with the most discrimination capacity did not reduce the overall classification ability of our model. This fact is likely to facilitate implementation in clinical practice. Data from other studies have shown better diagnostic estimators for several of the clinical features that were included in this study, either in isolation or when integrated within multivariable models [21,22,23]. Estimators of precision well over 85% and ROC AUCs of 0.85–0.91 have been previously reported in the literature, findings that contrast with the more conservative AUC of roughly 0.75 that was found in both our simplified and full multivariable model [24,25,26]. It is likely that methodological differences explain these contrasts in accuracy, such as the retrospective nature of the aforementioned studies and the inclusion of non-clinical variables within models, such as computer tomography scans or laboratory data; in addition, the non-use of protective means may increase the predictive properties of the model. However, it seems less likely that general laboratory information could result in an increase in diagnostic accuracy given the results from a recent systematic review [27]. We sought to develop a model that could aid in the diagnosis of COVID-19 in settings of restricted access to RT-PCR. Given the nature of the pandemic, it is likely that restrictions to other laboratory tests and imaging modalities could happen as well and, therefore, we chose not to add these sources of information to our calculations.

This study found associations with several demographic factors that modified the probability of COVID-19. The most relevant risk factor was a history of having a close contact with a confirmed infection by SARS-CoV-2, which had an adjusted odds ratio of 2.27 (95% CI 2.24–2.30, *p* < 0.001). This finding is in agreement with previous studies on the subject [28] and could be of even greater importance with the emergence of newer variants, such as omicron [29]. Having a private insurance provider decreased the probability of a SARS-CoV-2 infection, while identifying oneself as belonging to an indigenous Chilean population increased the risk of COVID-19. Rather than representing independent risk factors, it is likely that these associations stem from their link with other socioeconomic determinants of health. In Chile, persons with higher income tend to receive healthcare in the private sector, a well-known fact that has led to the use of this variable as a proxy of socioeconomic status in Chile [30]. On the other hand, native Chileans tend to have a lower income [31,32], which in turn translates into living conditions that facilitate viral spread, such as overcrowding, less access to barrier equipment and, a reduced capacity to comply with quarantines due to greater economic needs.

## 5. Strengths and Limitations

One of the most important strengths of this study is its sample size, which was achieved thanks to a concerted effort of the Chilean Ministry of Health and clinical researchers to improve the quality of epidemiological surveillance using a centralised system named EPIVIGILA. Given the nationwide nature of the data, a representative sample of Chilean patients has been obtained, including participants from different socioeconomic status, native Chileans, and travellers. This sample size has allowed the conduction of multivariable models with sufficient power to also integrate key epidemiological features that can also modify the probability of an infection by SARS-CoV-2. However, there are also several limitations that need to be taken into account as well. The most important one stems from the nature of the data. Although a national registry whose completion is enforced by law seems like a reliable source of information, the possibility of missing data cannot be fully excluded. It is possible that registries completed by physicians working in settings with a high demand for medical care omitted relevant data. Clinical features were reported by the patient, thus adding a subjective component that is inherent to self-reported data. No standardised procedure was established to gather information, which could also hamper the reliability of some key features such as dysgeusia or anosmia, in which some elements of the physical examination might increase their overall accuracy. In cases where obtaining a detailed history was not possible, such as cognitive decline or a severe respiratory failure, data was gathered from a proxy, which adds uncertainty to the assessment. Another relevant consideration is in the use of RT-PCR as the sole reference standard to confirm COVID-19 cases. Although the use of this test has been widely enforced by clinical guidelines, its diagnostic accuracy is suboptimal for a reference standard, with relatively low sensitivity that increases as the disease progresses, peaking roughly during the fifth day of symptoms.

## 6. Conclusions

No single clinical feature can fully confirm or exclude an infection by SARS-CoV-2. However, the combination of several demographic and clinical factors had a fair to high diagnostic accuracy in identifying patients with the disease. This model can help clinicians tailor the probability of COVID-19 and select diagnostic tests appropriate to their setting. The properties of the built model can be further validated in a recent set of PCR-tested patients.

## Figures and Tables

**Figure 1 biology-11-01136-f001:**
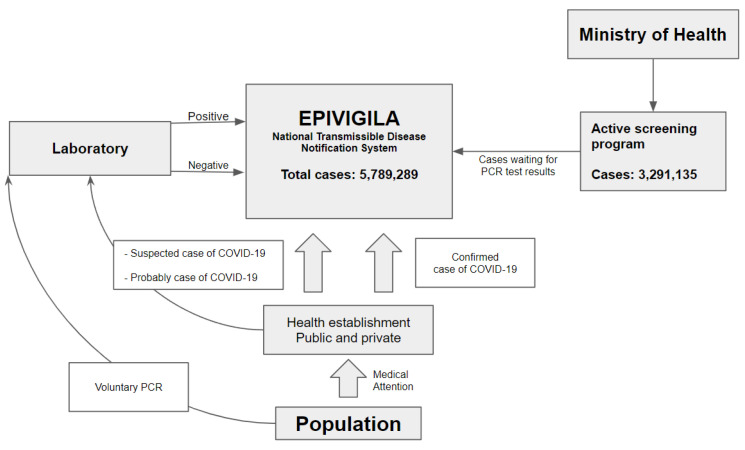
Epivigila is the national system to notify transmissible diseases. COVID-19 cases come in 3 ways: (1) health establishment (public and private) through health attention (confirmed, suspect, and probable); (2) laboratory: people that voluntarily realize the PCR; (3) active screening program: active search for cases in the community. (1) Suspected and probable cases in health facilities are registered on the platform and once the laboratory results (PCR) are available, they are updated; (2) the laboratory results of patients who voluntarily underwent the PCR are sent to Epivigila; (3) “active screening program” cases are recorded in Epivigila and updated with the results of the PCR examination.

**Figure 2 biology-11-01136-f002:**
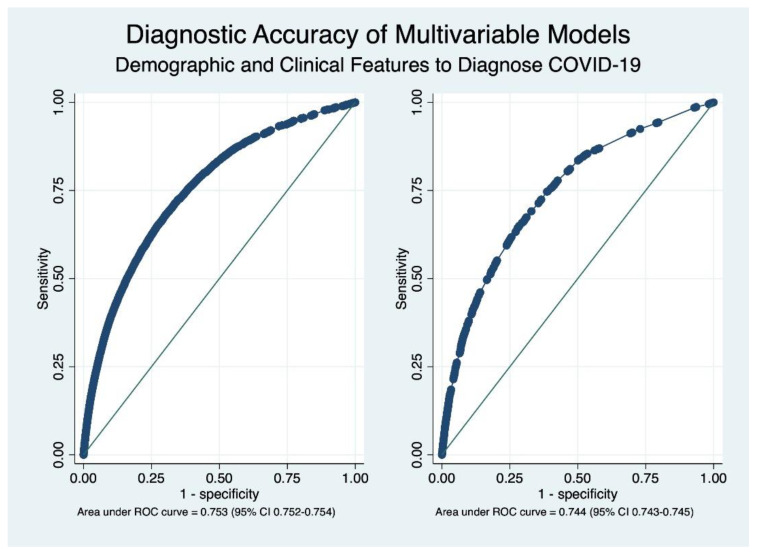
Receiver operating characteristics (ROC) curves depicting the overall diagnostic accuracy of complete (**left**) and simplified (**right**) models. As shown in the graph, the overall diagnostic accuracy is very similar.

**Table 1 biology-11-01136-t001:** Participant characteristics.

Characteristic	Persons without COVID-19 (*n* = 1,889,890)	Persons with COVID-19 (*n* = 297,180)	Total (*n* = 2,187,070)	*p*-Value
Mean Age (years, SD)	43.0 ± 17.5	43.8 ± 17.0	43.1 ± 17.5	<0.001 ^1^
Female Gender (%)	52.8%	49.7%	52.4%	<0.001 ^2^
Nationality (%)				
Chilean	92.1%	91.4%	92.0%	<0.001 ^2^
Not-Chilean	7.9%	8.6%	8.0%	
Indigenous Chilean (%)	2.5%	3.9%	2.7%	<0.001 ^2^
Site of Residence (%)				
Metropolitan Region	40.2%	37.4%	39.8%	<0.001 ^2^
Northern Chile	8.9%	12.4%	9.3%	
Central Chile	10.4%	9.3%	10.2%	
South-Central Chile	24.5%	24.2%	24.4%	
Southern Chile	13.6%	12.4%	13.4%	
Austral Chile	2.6%	4.3%	2.8%	
Health Insurance (%)				
Public (FONASA)	75.1%	77.8%	75.5%	<0.001 ^2^
Private (ISAPRE)	19.6%	15.3%	19.0%	
Other	5.3%	6.9%	5.5%	
Arterial Hypertension (%)	11.2%	15.2%	11.7%	<0.001 ^2^
Diabetes Mellitus (%)	5.5%	8.3%	5.9%	<0.001 ^2^
Asthma (%)	2.8%	2.8%	2.8%	0.675 ^2^
Cardiovascular Disease (%)	1.1%	1.0%	1.1%	<0.001 ^2^
Immunosupression (%)	0.84%	0.62%	0.81%	<0.001 ^2^
Liver Disease (%)	0.30%	0.21%	0.29%	<0.001 ^2^
Kidney Disease (%)	1.0%	0.97%	1.01%	0.013 ^2^
Chronic Lung Disease (%)	1.5%	1.2%	1.5%	<0.001 ^2^
Chronic Neurologic Disease (%)	0.64%	0.51%	0.62%	<0.001 ^2^
Suspected Contact (%)	2.01%	2.29%	2.05%	<0.001 ^2^
Confirmed Contact (%)	9.32%	18.3%	10.5%	<0.001 ^2^
International Travel (%)	1.3%	0.28%	1.18%	<0.001 ^2^
National Travel	0.81%	0.35%	0.75%	<0.001 ^2^

FONASA: Fondo Nacional de Salud. ISAPRE: Institución de Salud Previsional; ^1^ Student’s *t*-Test; ^2^ Pearson Chi^2^.

**Table 2 biology-11-01136-t002:** Diagnostic accuracy of clinical manifestations.

Clinical Feature	Prevalence	Sensitivity (95% CI)	Specificity (95% CI)	LR (+) (95% CI)	LR (−) (95% CI)	DOR (95% CI)
Headache	39.0%	56.5% (56.4–56.7%)	63.8% (63.7–63.8%)	1.56 (1.55–1.57)	0.60 (0.601–0.602)	2.30 (2.27–2.31)
Myalgia	32.7%	53.3% (53.1–53.5%)	70.6% (70.5–70.6%)	1.81 (1.80–1.82)	0.66 (0.659–0.664)	2.74 (2.71–2.76)
Cough	31.6%	51.1% (51.0–51.3%	71.5% (71.4–71.5%	1.79 (1.78–1.80)	0.684 (0.681–0.686)	2.62 (2.60–2.64)
Sore Throat	25.7%	33.2% (33.0–33.4%)	75.5% (75.4–75.5%)	1.35 (1.35–1.36)	0.885 (0.883–0.888)	1.53 (1.52–1.54)
Fever	15.5%	30.6% (30.4–30.7%)	86.9% (86.9–87.0%)	2.34 (2.32–2.35)	0.80 (0.799–0.801)	2.34 (2.32–2.35)
Diarrhea	8.9%	9.18% (9.07–9.28%)	91.0% (91.0–91.1%)	1.02 (1.01–1.04)	0.998 (0.996–0.999)	1.03 (1.01–1.04)
Dyspnea	8.7%	11.1% (11–11.2%)	91.6% (91.6–91.7%)	1.33 (1.31–1.34)	0.97 (0.960–0.972)	1.37 (1.35–1.38)
Abdominal Pain	7.0%	5.66% (5.58–5.75%)	92.8% (92.7–92.8%)	0.783 (0.77–0.795)	1.02 (1.02–1.02)	0.77 (0.757–0.782)
Chest Pain	5.1%	7.15% (7.05–7.24%)	95.2% (95.1–95.2%)	1.48 (1.46–1.50)	0.976 (0.975–0.977)	1.52 (1.50–1.54)
Anosmia	5.0%	17.7% (17.6–17.9%)	97.0% (97.0–97.0%)	5.89 (5.83–5.96)	0.85 (0.847–0.850)	6.95 (6.86–7.04)
Dysgeusia/Ageusia	4.1%	13.9% (13.8–14.0%)	97.5% (97.4–97.5%	5.47 (5.44–5.54)	0.883 (0.882–0.885)	6.19 (6.11–6.28)
Tachypnea	1.2%	1.71% (1.66–1.75%)	98.9% (98.9–98.9%)	1.55 (1.51–1.60)	0.994 (0.993–0.994)	1.56 (1.51–1.61)
Prostration	0.4%	0.59% (0.56–0.61%)	99.5% (99.5–99.5%)	1.24 (1.18–1.3)	0.99 (0.99–0.99)	1.24 (1.18–1.31)
Cyanosis	0.16%	0.23% (0.21–0.25%)	99.9% (99.8–99.9%	1.57 (1.44–1.7)	0.999 (0.999–0.999)	1.57 (1.44–1.70)

LR: likelihood ratio; DOR: diagnostic odds ratio.

**Table 3 biology-11-01136-t003:** Multivariable logistic regression: complete model.

Characteristic	Adjusted Odds Ratio (aOR)	95% CI	*p*-Value
Demographic Characteristics			
Male Sex	1.21	1.20–1.22	<0.001
Age > 65 Years	1.18	1.16–1.19	<0.001
Site of Residence			
Northern Chile	1.52	1.50–1.54	<0.001
Central Chile	0.82	0.81–0.83	<0.001
South-Central Chile	1.08	1.07–1.09	<0.001
Southern Chile	0.98	0.97–0.99	0.022
Austral Chile	2.08	2.04–2.13	<0.001
Private Insurance	0.90	0.89–0.91	<0.001
Indigenous Chilean	1.44	1.41–0.148	<0.001
Suspected Contact	1.24	1.21–1.28	<0.001
Confirmed Contact	2.27	2.24–2.30	<0.001
Clinical Symptoms			
Headache	1.29	1.28–1.30	<0.001
Myalgia	1.68	1.67–1.70	<0.001
Dyspnea	0.96	0.95–0.98	<0.001
Anosmia	3.72	3.66–3.79	<0.001
Dysgeusia/Ageusia	1.81	1.77–1.84	<0.001
Cough	1.88	1.86–1.90	<0.001
Fever	2.23	2.21–2.26	<0.001
Chest Pain	0.97	0.95–0.98	<0.001
Sore Throat	0.87	0.86–0.88	<0.001
Abdominal Pain	0.61	0.60–0.62	<0.001
Prostration	0.97	0.91–1.02	0.30
Diarrhea	0.75	0.74–0.76	<0.001
Tachypnea	1.25	1.21–1.29	<0.001
Cyanosis	1.37	1.25–1.50	<0.001
Constant	0.056	0.056–0.057	<0.001

CI: confidence interval.

**Table 4 biology-11-01136-t004:** Multivariable logistic regression: simplified model.

Characteristic	Adjusted Odds Ratio (aOR)	95% CI	*p*-Value
Demographic Characteristics			
Male Sex	1.21	1.20–1.22	<0.001
Age > 65 Years	1.13	1.11–1.14	<0.001
Indigenous Chilean	1.52	1.48–1.56	<0.001
Private Insurance (ISAPRE)	0.91	0.90–0.92	<0.001
Confirmed Contact	2.30	2.28–2.33	<0.001
Clinical Symptoms			
Myalgia	1.78	1.76–1.80	<0.001
Anosmia	3.80	3.72–3.86	<0.001
Dysgeusia/Ageusia	1.80	1.76–1.83	<0.001
Cough	1.86	1.85–1.88	<0.001
Fever	2.23	2.20–2.25	<0.001
Abdominal Pain	0.54	0.53–0.55	<0.001
Constant	0.06	0.06–0.062	<0.001

CI: confidence interval.

## Data Availability

The data used in this study are available on request made to the Chilean Ministry of Health.

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
