# Peer review of "Sensitivity and Specificity of Patient-Reported Clinical Manifestations to Diagnose COVID-19 in Adults from a National Database in Chile: A Cross-Sectional Study"

_biology, 2022, doi:10.3390/biology11081136_

Round 1

Reviewer 1 Report

The work is well planned, it is a prospective observational study. The methodology is well described and the foundations are correct. Likewise, the discussion and conclusions agree with the results, and the authors have reflected on the multiple limitations of the study.

Author Response

Thank you for your comments. We have included additional references to our manuscript and updated others in order to improve its quality.

Reviewer 2 Report

The paper assess the diagnostic accuracy of patient reported clinical manifestations to identify cases of COVID-19. The paper uses the logistic regression technique as a predictive model to examine the objective.

My specific comments:

(1) In a statistical analyses section - the authors use the level of significance for the model selection as 15%. This is very unusual in practice. They also further state that all candidate models considered the possibility of interactions between independent variables during their construction. I do not see any such interactions. Does it mean that all the interactions are statistically significant?

(2) In table 1: The number of persons with and without COVID do not add to total.

(3) The data for the Age in the logit regression model is transformed into a binary variable with a threshold level of 65. Why don't the authors use the Age variable as it is and also include the square of Age to capture the degree of non-linearities.

(4) Most of the symptoms (described as a binary independent variables) are likely to occur simultaneously. It leads to a collinearity problem. How do you overcome such problem? 

(5) The most serious problem is a endogeneity problem. Most of the symptoms are likely to be driven by the COVID, but not vice versa. In such case, the estimates will yield biased results and hence one will not be able access the reliability of the results. 

(6) The model accuracy can be established through a hold-out sample. The author should evaluate the accuracy carefully for both COVID and non-COVID cases.

(7) The authors should interpret the regression coefficients and provide the economic significance of these results. Many coefficients are less than 1 in the logit model. What does it imply in logistic regression framework.

Author Response

Thank you very much for your time reviewing our manuscript. I hope our answer addresses all of your concerns.

  1. The statistical model strategy is to produce sensibility, specificity, and predictive positive and negative values, and therefore interaction makes no sense in this strategy. Interactions are always valid in situations in which investigators use a statistical model to find risk factors for one outcome, and look for eventual effector modification of the control variables on the exposure.
  2. We corrected the number of person in table 1. 
  3. Age was dichotomized at 65 since is the cutoff point for older adults.
  4. We do not detect any collinearity in our data.
  5. The objective of our study is to assess the diagnostic accuracy for COVID-19 cases (PCR tested) based on patient-reported clinical manifestations, we used symptoms as associated factors for positive cases. Is or understanding that endogeneity is not a problem as manifestations are not being considered to be causal of Sars-Cov2 infection.
  6. One of the strategies for building predictive models is to use the split-sample method, as you commented. Another strategy is to validate the model by computing sensibility, specificity, and predictive values (positive and negative), in addition to the area under the ROC curve, which is the strategy we decided to use.
  7. We agree with the reviewer that conducting an economic analysis of our findings would be of great interest to our results. However, this was not part of our primary objectives, and thus no particular methodology was prespecified to conduct these analyses. Therefore, we believe that it would not be appropriate to include these elements within our manuscript, but we hope to obtain support and authorisations from our review board to include this aspect in another article in the future.

Reviewer 3 Report

While the manuscript is well-written, the following need to taken care of: 

1. It will be nice if the abstract is rewritten to summarize the methodology and the results, emphasizing on what is novel.

2. Separate citation numbers from the rest of the text. 

3. Line 123: Change Chi2 to Chi Square of use the symbol. 

Author Response

Thank you very much for your time reviewing our manuscript. As suggested, we have made several changes to our text to address your concerns. 

  1. A simple summary has been added to the study in order to highlight its findings.
  2. As requested by the reviewer, we have separated the citation numbers in parentheses from words throughout the manuscript.
  3. Chi2 now appears written as "Chi Square"

Reviewer 4 Report

Dear Authors

Thank you very much for your manuscript submission. Your study is very interesting. However, it is recommended to show all the mentioned items both in male and female patients; And then compare them with each other. The male patients are bold in your manuscript. What about the female patients?

In this regard, all the sections of the manuscript should be revised.

It is recommended to increase the number of the references of the manuscript to support your data and information.

Moreover, it is recommended to add a flow chart to show the employed procedures in the present study to the readers at one glance. Please download a flow chart from CmapTools (https://cmap.ihmc.us) for free as a material from the following paper. In this regard, please do read and add the following paper to the References section of the manuscript:

Writing a strong scientific paper in medicine and the biomedical sciences: a checklist and recommendations for early career researchers. Biol Futur. 2021 Dec;72(4):395-407. doi: 10.1007/s42977-021-00095-z. Epub 2021 Jul 28. PMID: 34554491.

Figure 1 needs a revised legend

Author Response

Thank you very much for your interest in our study. As suggested, several changes have been made to our text to address most of your concerns. Please find our response and reasons why some suggestions were not implemented  below:

  1. The frequency of male participants was chosen as a descriptor regarding biologic sex solely because its frequency was slightly greater than the frequency of female participants. We were not able to find a section in which the word “male” was highlighted using bold, which might be a software issue. We have removed the bold format of the word altogether throughout the text. 
  2. This study did not seek to compare differences between the sexes regarding clinical manifestations of COVID-19. Such an analysis would be lengthy, and likely result in a major switch to our aim, which was to provide clinicians with a tool that could help tailoring the need of diagnostic tests during the COVID-19 pandemic. It should also be considered that biologic sex was one of the epidemiological variables that was included in all our models. Therefore, we respectfully decline to add this analysis to our manuscript.
  3. As suggested by the reviewer, we have increased the number of references to 30.
  4. We have added a revised legend to Figure 1 explaining its implications.

Round 2

Reviewer 2 Report

I do not see that the authors have taken any of my statistical comments seriously. They have disposed all comments except fixing the typo in the sample size. I do not agree with their response. 

The model severely  suffer from endogeneity problem. Failure to address the problem leads to biased estimates. 

Unfortunately, I can not recommend for publication knowing that the results based on the statistical analysis are biased and inconsistent. 

Author Response

We apologise with the reviewer if any of our responses led him/her to believe that his/her insights were unimportant to our research. This was not intended in our response. 

Logistic regression analyses have been used previously as modelling techniques to assess an overall probability of disease. Several other researchers have used this approach to construct models to establish a pre-test probability of a myriad of diseases, including COVID-19(1,2), appendicitis(3) and seizures(4), amidst many others. 

We understand endogeneity as a term primarily used in econometric analyses that describes a phenomenon in which the value of an independent variable is dependent on the value of another predictor variable(5). Because of this endogeneity, significant correlation can exist between factors contributing to both the endogenous independent variable and the dependent variable, which results in biased estimations of correlation coefficients. This can result in significant multicollinearity and interactions between variables, of which no evidence was found within our dataset during the development phases of our models.

In addition, endogeneity results in model-generated coefficient standard errors that are much larger than true standard errors, thus hampering the statistical power of the study. The net effect of this source of bias would then be the occurrence of missed associations between clinical features, epidemiological risk factors and receiving a diagnosis of COVID-19. However, it should be kept in mind that our sample size greatly exceeds that of previous studies due to the nationwide nature of our dataset, thus making it unlikely for a relevant association to be missed in our study. It should also be kept in mind that a higher threshold for association was used for detecting potentially relevant variables during model development, which was aimed at reducing the probability of a type-II error. 

We hope that these arguments help answer the reviewer’s concerns.  

References

  1. Saegerman C, Gilbert A, Donneau AF, Gangolf M, Diep AN, Meex C, et al. Clinical decision support tool for diagnosis of COVID-19 in hospitals. PLoS One. 2021 Mar 11;16(3):e0247773.
  2. Romero-Gameros CA, Colin-Martínez T, Waizel-Haiat S, Vargas-Ortega G, Ferat-Osorio E, Guerrero-Paz JA, et al. Diagnostic accuracy of symptoms as a diagnostic tool for SARS-CoV 2 infection: a cross-sectional study in a cohort of 2,173 patients. BMC Infect Dis. 2021 Mar 11;21(1):255.
  3. Samuel M. Pediatric appendicitis score. J Pediatr Surg. 2002 Jun;37(6):877–81.
  4. Sheldon R, Rose S, Ritchie D, Connolly SJ, Koshman ML, Lee MA, et al. Historical criteria that distinguish syncope from seizures. J Am Coll Cardiol. 2002 Jul 3;40(1):142–8.
  5. Avery G. Endogeneity in logistic regression models. Emerg Infect Dis. 2005 Mar;11(3):503–4; author reply 504–5.

Reviewer 4 Report

Dear Authors

Thank you for your clear explanation and revision.

1. In your explanation you have mentioned "The frequency of male participants was chosen as a descriptor regarding biologic sex solely because its frequency was slightly greater than the frequency of female participants."

As you know in an interesting work like yours both of male and female patients should be involved. In this regard, it is recommended to revise this part of your manuscript.

2. It is recommended to read and add the following paper to Reference section of your manuscript to have fruitful Introduction and Discussion section:

The Global Emergency of Novel Coronavirus (SARS-CoV-2): An Update of the Current Status and Forecasting. Int J Environ Res Public Health. 2020 Aug 5;17(16):5648. doi: 10.3390/ijerph17165648. PMID: 32764417; PMCID: PMC7459861.

3. It is recommended to add a flow chart to show the employed procedures in the present study to the readers at one glance. Please download a flow chart from CmapTools (https://cmap.ihmc.us) for free as a material from the following paper. In this regard, please do read and add the following paper to the References section of the manuscript:

Writing a strong scientific paper in medicine and the biomedical sciences: a checklist and recommendations for early career researchers. Biol Futur. 2021 Dec;72(4):395-407. doi: 10.1007/s42977-021-00095-z. Epub 2021 Jul 28. PMID: 34554491.

4. According to aforementioned comments, it is recommended to revise Discussion section too.

5. I believe that an interesting work like yours, deserves a prestigious Conclusion section. So, Conclusion should be enriched and revised.

Author Response

1.- The sample is composed of XX% of male and YY% of female.

2.- We add the proposed reference.

3.- We add a flow chart to our article.

4-5. We made the proposed changes.